# Relationship between Internet Addiction, Personality Factors, and Emotional Distress among Adolescents in Malaysia

**DOI:** 10.3390/children9121883

**Published:** 2022-12-01

**Authors:** Soh Chou Fu, Nicholas Tze Ping Pang, Walton Wider

**Affiliations:** 1Department of Psychiatry, Faculty of Medicine and Health Science, Universiti Malaysia Sabah, Kota Kinabalu 88400, Sabah, Malaysia; 2Faculty of Business and Communications, INTI International University, Nilai 71800, Negeri Sembilan, Malaysia

**Keywords:** Internet addiction, emotional distress, personality, COVID-19, emotional stability

## Abstract

COVID-19 has significantly affected the mental health of adolescents, thus increasing the emotional distress among them. Studies have reported that heavy Internet use during COVID-19 was linked with poor mental health among adolescents. Additionally, it was found that personality factors are linked with mental health in general. Although past literature has reported the effect of personality factors on mental health, there are limited studies examining the underlying mechanisms among Malaysian adolescents. Therefore, the current study offers an understanding of the intervening role of personality factors in the relationship between Internet addiction and emotional distress among adolescents in Malaysia. This study also aimed to determine the prevalence of Internet addiction and emotional distress (depression, stress, and anxiety) among adolescents in Malaysia. There are a total of 500 participants from 7 secondary schools who range from 13 to 19 years of age. This was a cross-sectional study, and 3 valid questionnaires were used: The Internet Addiction Test (IAT), the Depression, Anxiety, and Stress Scale (DASS-21), and the Ten-Item Personality Inventory (TIPI). Partial least square structure equation modelling (SmartPLS) was used to analyse the mediation models. The results showed that the prevalence of Internet addiction among adolescents was 29.6% and the prevalence of depression, anxiety, and stress among adolescents were 64.8%, 78%, and 51.4%, respectively. Furthermore, mediation analysis showed only emotional stability mediated the relationship between Internet addiction and emotional distress, but not openness to experience, extraversion, conscientiousness, or agreeableness. It is proposed that mental health providers should focus on providing emotion-related interventions to adolescents confronting COVID-19 challenges, ultimately improving mental health.

## 1. Introduction

According to the United Nations International Children’s Emergency Fund’s (UNICEF) latest flagship report, “The States of the World’s Children”, which is focusing on the mental health of children and adolescents for the first time in history, there is a rising trend of mental health issues among children and adolescents [1]. It is estimated that between the years 2016 and 2019, in the United States, approximately one in ten children will be diagnosed with a form of anxiety, and approximately one in twenty children will be given the diagnosis of depression [2]. This problem is made even worse by the pandemic COVID-19, as the most recent data shows that one in three secondary school students experience poor mental health and that half of them persistently feel hopeless and sad during the COVID-19-era [3]. In one study in Palestine showed that 89.1%, 72.1%, and 35.7% of secondary school students experienced moderate to severe levels of anxiety, depression, and stress, respectively [4]. Other than that, there is one meta-analysis conducted in China during COVID-19 pandemic demonstrated that the prevalence of stress, depression, and anxiety are 48.1%, 26.8%, and 31.8%, respectively [5]. This is an issue that was already a problem, but the pandemic COVID-19 has made it even worse. In order to stop the further spread of COVID-19, people are being warned all over the world to keep a safe distance from one another, stay away from crowded areas, and scale back on their travelling. People will spend a significant amount of their time playing electronic gadgets, which is likely to cause behavioural addiction in the younger generation, in particular [6], despite the fact that the source of happiness is limited to activities that can be conducted alone or inside where it is quiet. On the other hand, as the Internet has become an integral part of many people’s lives, there has been an increase in the number of cases of Internet addiction. This trend is expected to continue. During the time of the COVID-19 pandemic, more than one third of the population of China struggled with addiction to the Internet [7]. Besides that, there are two studies conducted in Taiwan and Iran during COVID-19 pandemic showed that prevalence of Internet addiction among secondary school students are 24.6% [8] and 31.53% [9], respectively. These are relatively high prevalence rates when compared to the United States and Europe; which, prior to the era of COVID-19 had a prevalence rate that ranged from 1.5% to 8.2% [10]. Additionally, the rate of addiction to the Internet among adolescents is significantly higher than that of adults [11].

Numerous studies have been conducted, and the results show that there is a direct relationship between Internet addiction and emotional distress, particularly depression, and anxiety [12,13,14]. Because of the COVID-19 pandemic and the devastating effects it has had on our population, particularly the younger generation, this is an important area for research because both addiction to the Internet and mental health problems are showing signs of worsening in recent years [15]. In addition to this, there is a substantial body of evidence that demonstrates relationships between excessive internet use and factors of personality. One meta-analysis was carried out in 2016 and found that a positive correlation exists between neuroticism and Internet addiction [16]. On the other hand, a negative correlation exists between Internet addiction and openness to new experiences, conscientiousness, extraversion, and agreeableness. Research carried out in India, China, and Malaysia [17,18,19] lends further credence to the findings presented here. However, the vast majority of the evidence supporting the relationship between them was gathered before COVID-19, and as a result, it is of the utmost importance to re-establish the evidence during the time period covered by COVID-19. As a result, the significance of this study cannot be mitigated because numerous studies have shown that an addiction to the Internet is positively correlated to emotional distress [20]. However, there is currently a paucity of research that investigates the underlying mechanism that may potentially interfere the relationship between an Internet addiction and emotional distress amid COVID-19 [21]. Chaplin suggested a mediator and moderator model to comprehend the possible roles performed by personality in a connection, namely whether personality is a mediator or moderator of the interaction between variables [22]. The mediator and moderator models are conceptually comparable to essentialist and contextualist viewpoints on personality [23]. The essentialist view on personality asserts that personality is a hereditary feature that is stable and difficult to alter; hence, personality is more likely to play the function of a moderator; consequently, research adopting this approach focus on how various personalities interact with one another or how personality and circumstance interact with one another to produce certain psychological consequences. In contrast, the contextualist perspective on personality asserts that personality is a state that can be altered due to rapid physical, cognitive, and social changes; therefore, personality is more likely to play the role of a mediator, and studies that adopt this perspective focus on the question of why a relationship exists [22,24,25]. On the basis of this viewpoint, we used the contextualist approach and hypothesised that Internet addiction among adolescents would influence their personality traits and, in turn, their emotional distress.

Hence, this study aims to identify the underlying personality traits that lead to Internet addiction behaviour and emotional distress symptoms. In addition, this study also aims to determine the level of emotional distress and addiction to the Internet that is prevalent among adolescents in Malaysia. In this regard, the construction of the mediation model and prevalence study will not only facilitate an understanding of the relationships between these variables, but will also have clinical relevance by contributing to the development of potential interventions to improve the mental health of adolescent affected by the epidemic.

## 2. Materials and Methods

### 2.1. Participants

In this particular study with a cross-sectional design, participants included five hundred secondary school students who had been chosen using homogeneous convenience sampling from seven different schools located in Sabah and Johor, Malaysia. Because it is known that traditional convenience sampling methodology is less generalisable and accurate than a homogeneous convenience sampling strategy, which can lead to estimation bias, a homogeneous sampling strategy was adopted in this study [26]. This is because estimation bias can be caused by traditional convenience sampling methodology. The homogeneous convenience sampling method ensured that participants were high school students between the ages of 13 and 19, that they were capable of comprehending the questionnaire, and that they did not suffer from a severe mental or physical illness at the time the investigation was being conducted. We utilised the G*Power analysis to determine the minimal sample size necessary to achieve statistical power [27]. This study’s model had five predictors. Using G*Power with an effect size of 0.15, alpha of 0.05, and power of 0.95, the minimal number of samples required was 138. Thus, we can confidently assert that our study with a sample size of 500 has a power of greater than 0.95 and is sufficiently large, and that the findings may be used with assurance. This study was carried out in accordance with the criteria for Strengthening the Reporting of Observational Studies in Epidemiology (STROBE) [28].

### 2.2. Questionnaires

The following are the four sub-sections that make up this questionnaire: the sociodemographic questionnaire; the Internet Addiction Test; the Depression, Anxiety, and Stress Scale; and the Ten Item Personality Inventory.

#### 2.2.1. Sociodemographic Questionnaire

This was a brief survey that asked for demographic information such as age, gender, race, and total median household income. There are three sub-categories for the total median household income, and they are labelled B40 (less than RM4850), M40 (between RM4850 and RM10959), and T20 (more than RM10960) [29].

#### 2.2.2. Internet Addiction Test (IAT)

Young [30] was the first person to create the IAT. It has been localised into a great number of languages and is the instrument that is utilised the most frequently for gaining access to one’s dependence on the Internet. A Likert scale with five points is used to rate each of the 20 items in it. A normal level of Internet addiction is indicated by a score between 0 and 30, a mild level by 31 to 49, a moderate level by 50 to 79, and a severe level by 80 to 100. The six factors that were developed had positive correlations to each other with ranges that went from 0.226 to 0.62 [31]. The English version of the IAT has good internal consistency, as measured by a Cronbach alpha that falls somewhere in the range of 0.54 to 0.82. Furthermore, the concurrent validity was acceptable [32].

#### 2.2.3. Depression, Anxiety, and Stress Scale (DASS-21)

The Distress and Anxiety Scale for Children (DASS-21) is a well-established self-report scale that was developed by Lovibond and Lovibond [33] to measure the level of emotional distress (depression, anxiety, and stress) in children. This scale has 21 items, each of which is rated on a four-point scale, ranging from 0 to 3. In order to calculate the final score, the individual domain scores will need to be added together and then multiplied by two. The minimum acceptable score for depression is 9, the minimum acceptable score for anxiety is 14, and the minimum acceptable score for stress is 14. The severity of emotional distress is ranked from mild to extremely severe, and higher scores in each domain indicate a greater degree of severity of emotional distress in that domain. The psychometric properties were validated with a satisfactory Cronbach’s alpha of 0.94, 0.87, and 0.91, respectively, for depression, anxiety, and stress. Additionally, the concurrent validity was satisfactory [34].

#### 2.2.4. Ten Item Personality Measure (TIPI)

Gosling [35] developed the short measure known as the TIPI in order to gain insight into the personality characteristics of an individual. The questionnaire consists of ten questions, each of which is rated on a 10-point scale ranging from 1 (strongly disagree) to 7 (strongly agree). Two questions are devoted to assessing each of the Big Five personality traits: extraversion, agreeableness, conscientiousness, emotional stability, and openness to experience. The positive pole is represented by one of the items in the dimensions, and the negative pole is represented by the other item in the dimensions. The Cronbach alpha for the TIPI falls in the moderate range, between 0.40 and 0.68. Both the test-retest reliability and the external correlation findings are satisfactory [34]. Several previous studies have indicated that the Cronbach’s alpha for the TIPI ranges from low to moderate, which is typical for shorter scales [36]. The goal of the TIPI was to be a concise and reliable instrument (including content validity). The production of an instrument with a high alpha and a good CFA fit was not the goal [35]. As a result, for the purpose of this investigation, the data analysis consisted of computing a numerical score based on the two items representing each personality trait.

### 2.3. Data Collection and Analyses

In consideration of the policy regarding the physical distancing of participants and in an effort to shield them from COVID-19, the research material was made available to all of the participants online in the form of a Google Form by the participants’ secondary school teachers. Before collecting data, the researchers visited teachers to discuss the administration of questionnaires. All teachers involved in the data collection method were briefed on how to explain the study’s aim and confidentiality, how to collect data, and how to respond to any inquiries from respondents. Teachers involved in the data collection method were also required to inform respondents that they could also refuse to respond to any question for any reason or withdraw from the study at any moment. The data was gathered beginning on the 13 May 2022 and continuing through the 28 May 2022. The surveys were completed by all respondents without any missing data instances. Using IBM SPSS version 27.0 (developed by Norman H. Nie, Dale H. Bent, and C. Hadlai Hull, Chicago, IL, USA), the sociodemographic data was analysed and described using descriptive statistics (including frequency, percentage, mean, and standard deviation). All of the test variables were evaluated for normality using kurtosis and skewness measurements, and the results indicated that distribution was normal. Skewness was anywhere from −2 to 2, and normal kurtosis was anywhere from −10 to 10 [37]. PLS-SEM was the approach that we used for the statistical analysis of the mediating model, and SmartPLS 3.3.3 was the software that we used for that particular purpose. PLS-SEM takes into account both the measurement model, also known as the outer model, and the structural model, also known as the inner model [38]. Indicator reliability and validity were analysed as part of measurement model assessment as a means of determining whether or not the measures that were used in this study were adequate. This was conducted so that the measurement model could be evaluated. The process entails determining whether or not each item can be relied on, determining whether or not each construct can be relied on for internal consistency reliability, and determining whether or not converging and discriminant validity can be established [39]. Regarding the evaluation of the structural model, an indirect effects analysis is carried out to provide an explanation of the mediating effect of personality factors between Internet addiction and emotional distress. The R-squared (R^2^) value, the significance of the indirect effects using the *p*-value and a confidence interval of 95 percent (CI 0.95), and the effect size (f^2^) must all be evaluated during the structural model assessment [39].

### 2.4. Reliability and Validity (Measurement Model Assessment)

In order to evaluate the structural and measurement models, we used a total of 500 samples. The first things to be tested for validity and reliability were the reflective constructs of Internet addiction and emotional distress. It is necessary to have an extracted average variance (AVE) that is greater than 0.5 and a composite reliability (CR) that is greater than 0.7 in order to demonstrate that convergent validity has been achieved [39]. Seventeen indicators had to be removed because there were not sufficient outer loadings. The results presented in Table 1 demonstrate that all reflective constructs have had their convergent validity and reliability verified. After that, the heterotrait-monotrait (HTMT) method was used to determine whether or not the discriminant validity was satisfactory. In order for the HTMT to have discriminant validity, the correlation between each pair of latent exogenous constructs must be lower than 0.85 [40]. The fact that all constructs in Table 2 have HTMT values that are lower than 0.85 provides evidence in support of the discriminant validity of the measures.

## 3. Results

### 3.1. General Characteristics of Participants

A total of 500 responses were received from students, all of which were usable. The majority of whom were Sabah natives (61%) and belonged to the B40 demographic group (77.4%). In total, 246 (49.2%) participants are male, and 254 (50.2%) participants are female. Their ages range from between 12 to 20 years old, with 15.5 years being the median age. Regarding school, majority of the participants were students of SMK Taman Tun Fuad (83.6%). The participants’ sociodemographic profiles are summarised in Table 3.

### 3.2. Results of the Questionnaires

The results of both the IAT and DASS-21 are presented in Table 4. On the IAT, the mean score was 39.57, and the standard deviation was 18.93. On the Internet Addiction Test, the prevalence of Internet addiction was found to be 29.6% (*n* = 148), with a cut-off score of 50, which corresponds to a “moderate level of Internet addiction”. Only 2% (*n* = 10) of the total number of participants who are addicted to the Internet are considered to have a severe level of addiction to the Internet. In terms of emotional distress, the mean scores and standard deviation for depression, anxiety, and stress are 14.87 (10.778), 16.15 (10.704), and 15.79 (10.187), respectively. In addition to that, 64.8%, 78%, and 51.4% of the participants reported feeling stressed, anxiety, or depressed, respectively. On the DASS-21 questionnaire, the cut-off points for depression, anxiety, and stress are scores of 10, 9, and 15, respectively, which correspond to the “mild level” of each condition.

### 3.3. Structural Model Assessment

We performed an analysis of the collinearity between the study variables to ensure that the structural model did not contain any problems caused by lateral collinearity [41]. According to Table 5, all of the inner VIF values were less than five, which suggests that the structural model did not have a significant issue with collinearity among the predictor constructs. We used a bootstrapping method with 5000 samples to determine the path coefficient (β), *t*-values, *p*-values, and R^2^ of the structural model [39]. This allowed us to test whether or not the indirect effect was present. Figure 1 depicted the R^2^ value, which ranged from 0.000 to 0.168 and accounted for 0.0% to 16.8% of the exploratory variance. In this particular investigation, the analysis of the mediator was carried out by the product of the coefficient approach with bootstrapping resampling [42]. According to the findings of our study, the only factor that acts as a mediator between Internet addiction and emotional distress is emotional stability (refer Table 5). Because of this, the overall mediating effect of emotional stability reflected the fact that addiction to the Internet had a negative relationship with emotional stability, which in turn had a negative relationship with emotional distress. The results of the indirect effect are listed in Table 5.

## 4. Discussion

The findings of this research indicate that a significant proportion of adolescent are addicted to the Internet (29.6%) which is higher compared to other studies conducted before the COVID-19 pandemic. There are two research projects conducted in China and India used the same cut-off scores and scale (IAT) to evaluate Internet addiction, the prevalence of Internet addiction among adolescents ranged from 6% to 20% [43,44,45]. Our findings also correspond to other research projects conducted during the COVID-19 pandemic [8,9]. The rising trend of prevalence of Internet addiction in younger generations needs immediate attention and action from various parties including government agency and regulatory authorities as Internet addiction may cause unwanted and undesirable consequences to our younger generations.

In term of emotional distress, the prevalence of depression (64.7%), anxiety (78%), and stress (57.4%) in our study is very high compared to previous studies conducted before the COVID-19 pandemic while it is consistent with the research conducted in Taiwan and Iran during the COVID-19 pandemic [4,5]. Before COVID-19, the prevalence of depression, anxiety, and stress among adolescents and young adults was found to range from 27 to 38%, 40 to 49%, and 27 to 35%, respectively [46,47,48]. Our study also showed that anxiety was the problem that was experienced the most frequently by secondary school students in our study. Considering the mental issue due to COVID-19, the levels seen in Malaysia were higher compared to China. The inference maybe premature as the numbers of studies conducted in Malaysia is few and it may take some time to see its full impact in Malaysia. However, we may face serious consequences if immediate actions are not taken to manage mental health. One of the serious and fearsome outcomes is suicidality, and there are 123 cases of COVID-19 related suicide and self-harm in Nepal reported by [49].

There have been a number of studies that have found a connection between addiction to the internet and emotional distress. These studies have found that the correlation is proportional, meaning that the higher the level of Internet addiction, the higher the score of emotional distress [50,51,52]. However, the primary objective of the current research was to investigate whether or not the Big Five personality traits play a role in mediating the connection between compulsive internet use and feelings of emotional distress. According to the current finding, the only factor that significantly mediates the relationship between Internet addiction and emotional distress is emotional stability. Emotional stability is an important individual personality trait that can help alleviate emotional distress and may protect a person from developing an addiction to the Internet. One of the possible explanations for this finding is that emotionally stable people recognise negative thoughts in a different way than those who are not emotionally stable. As a result, adolescents who are better at recognising negative thoughts, who are optimistic, and who frequently feel relaxed are more likely to be able to manage stress easily and less impulsively. There is a correlation between higher levels of emotional stability and lower average levels of emotional distress. This is due to the fact that people who are emotionally stable are better able to deal with stressful situations and better manage their emotions than people who are less emotionally stable. In point of fact, studies have shown that individuals with lower levels of emotional stability have stronger responses to stressful situations than individuals with higher levels of emotional stability [53,54]. In addition, Dodge’s [55] model of social information processing postulates that an individual’s perception of social and internal cues is a significant factor that plays a role in the way information is processed. Consequently, individuals whose interpretations of environmental cues are congruent with emotional stability may be more likely to participate in controlled, adaptive behaviour and experience less emotional distress. This is due to the fact that emotionally stable individuals may interpret environmental cues in a way that is consistent with their emotional stability. According to the findings of this research, having a steady emotional state appears to be one of the most important personality traits for warding off Internet addiction as well as emotional distress.

However, the possibility of reverse causality, such as the idea that emotional distress is an antecedent to Internet addiction itself, should be considered; this possibility was not taken into account in the current study. It is important to note that addiction to the Internet may increase the likelihood of experiencing emotional distress. Hartanto et al. asserted that there is a causal association that runs in the other direction between social media and depressed symptoms [56]. There appears to be a correlation between greater social media use and higher levels of depression. However, depressed symptoms may prompt greater use of social media, whether as a means of escape or for the purpose of reaffirming one’s sense of self-esteem through the approval of others. With regard to the findings of our study, experiencing emotional distress may come before developing an addiction to the Internet, and the symptoms of mental health problems may have an effect on whether or not an individual develops an addiction to the Internet. A person who is going through depression or stress could, for instance, use the Internet more often or for longer periods of time. In light of the fact that there is an imperative need to give scientific proof in this regard, undertaking research in the future that investigates this assumption might prove to be significantly important.

### Limitations, Implications, and Recommendations

Some strengths and limitations of the study should be noted. Firstly, it is a cross-sectional study, and hence causal inferences cannot be made. Nevertheless, cross-sectional study is best fit for our study as one of our aims is to find out the prevalence of Internet addiction and mental disorders. Second, this study consisted of a self-reported questionnaire, which may carry the possibility of recall bias. However, all subjects are given 2 weeks to answer the questionnaires, which give them enough time to recall and answer the questionnaire. Third, all the questionnaires were conveyed in English, which may have created a language barrier among the participants, who are mostly from secondary school. In order to overcome this, subjects are given clear instructions and guidance before answering questionnaires as to make sure they understand the questionnaires clearly. On the other hand, our study has participants equal in terms of gender. So, the results of the study can be used for both men and women without any bias. Lastly, we utilised the TIPI because we believed it to be an ideal instrument for internet research as asserted by [57]. TIPI gives evidence of measurement validity despite shortcomings in its internal consistency and reliability. However, due to the length of the Big Five scales, internal consistency may be an unreliable metric of the TIPI’s reliability, as Gosling et al. have remarked [35]. In light of this, it is recommended that future studies use a measurement that is both more reliable and has a higher degree of internal consistency.

Since emotional stability has been identified as one of the protective factors against emotional distress, emotion-focused intervention may also be incorporated into other forms of psychotherapy, such as mindfulness therapy, emotion-focused therapy, and cognitive-behavioural therapy. This is possible due to the fact that research has shown that these types of therapies are effective in regulating the emotions of adolescents [58]. Counsellors should also think about the possibility of incorporating skills for managing adolescents’ emotions into the counselling sessions they conduct. When adolescents are struggling to keep their mental health in good shape or have trouble breaking an addiction to the Internet, they should be taught how to practice mindfulness. When practiced by adolescents, mindfulness-based approaches have been shown to effectively lessen the intensity of negative emotional responses that are caused by intensifying psychiatric difficulties as well as exposure to stressors [59].

Some recommendations we suggest for future research are that the scope of the research should extend to adolescents from different states in Malaysia or involve adolescents from different countries. Consequently, the results should be interpreted with caution, as their limited applicability to adolescents necessitates future validation with other age groups. Other than that, we recommend developing a Malay-version of the personality questionnaire in order to eliminate the language barrier if research is to be conducted in Malaysia, as Malay is the official language in Malaysia. Besides that, the future field of research is recommended to focus on the development of a policy on the treatment of Internet addiction among adolescents. Aside from that, it is suggested that future researchers look into how the psychological services in Malaysia help adolescents who have problems with their mental health or are addicted to the Internet.

## 5. Conclusions

The purpose of this study was to investigate the relationship between Internet addiction, personality factors, and emotional distress, as well as the prevalence of Internet addiction and emotional distress among adolescents during the time period covered by the COVID-19 pandemic. The COVID-19 pandemic had a significant impact on Internet addiction and mental health issues, particularly among adolescents, making this study very important. The current study has a number of repercussions, both theoretical and practical, that can be drawn from it. Our research has shed light on the ways in which personality factors can act as a mediator between Internet addiction and emotional distress in adolescent populations in Malaysia. The findings also suggest that there was an increase in the levels of depression, anxiety, and stress among adolescents during the COVID-19 pandemic. The findings of this study can be used and applied by students, teachers, and researchers to develop strategies for coping with emotional stressors during the COVID-19 pandemic. This will allow for a better understanding of Internet addiction, personality factors, and emotional distress as well as a better overall understanding of these topics.

## Figures and Tables

**Figure 1 children-09-01883-f001:**
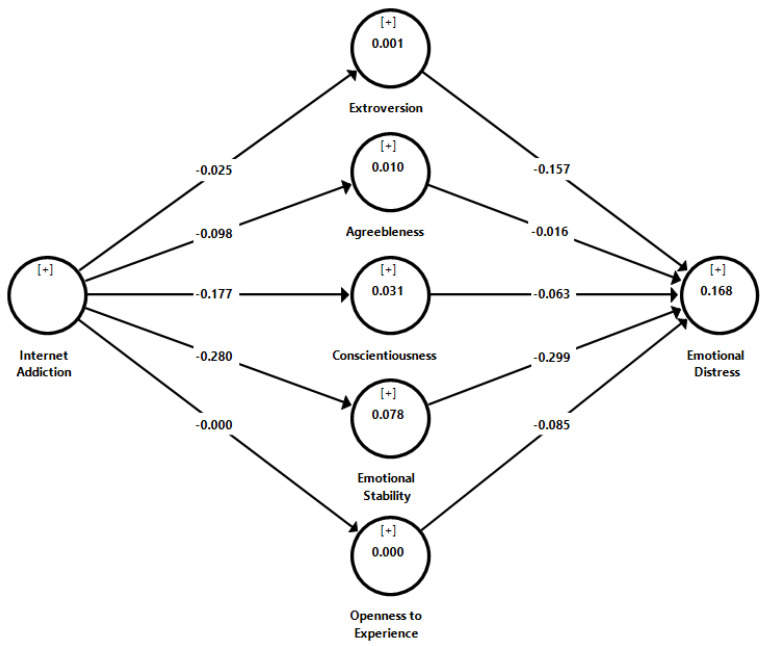
Results of structural model.

**Table 1 children-09-01883-t001:** Results of measurement model.

Construct	Items	Loadings	CR	AVE
Internet Addiction	IA20	0.784	0.877	0.504
	IA14	0.721		
	IA12	0.697		
	IA16	0.692		
	IA6	0.699		
	IA13	0.666		
	IA17	0.707		
Emotional Distress	A2	0.715	0.946	0.506
	A3	0.698		
	A4	0.682		
	A5	0.760		
	A7	0.733		
	D1	0.656		
	D2	0.666		
	D3	0.741		
	D4	0.711		
	D5	0.721		
	D6	0.740		
	D7	0.723		
	S2	0.695		
	S3	0.719		
	S4	0.767		
	S5	0.700		
	S6	0.653		

**Table 2 children-09-01883-t002:** Discriminant validity using HTMT ratio.

No	Constructs	1	2	3	4	5	6	7
1	Agreebleness							
2	Conscientiousness	0.309						
3	Emotional Distress	0.102	0.213					
4	Emotional Stability	0.227	0.316	0.363				
5	Extroversion	0.159	0.091	0.207	0.124			
6	Internet Addiction	0.106	0.177	0.495	0.274	0.054		
7	Openness to Experience	0.230	0.337	0.177	0.185	0.057	0.051	

**Table 3 children-09-01883-t003:** Participants’ sociodemographic profile.

Variables	Frequency (%)	Mean (SD)
Age		15.15 (1.65)
Sex Male Female	246 (49.2) 254 (50.8)	
Race Native Sabah Malay Chinese Indian	305 (61.0) 122 (24.4) 69 (13.8) 4 (0.8)	
Total Median Household Income B40 M40 T20	387 (77.4) 97 (19.4) 16 (3.2)	
Schools		
SMK Taman Tun Fuad	418 (83.6)	
SMK Kamarul Ariffin	31 (6.2)	
SMK Chaah	26 (5.2)	
Kota Kinabalu High School	16 (3.2)	
SMK Labis	8 (1.6)	
SMK Munshi Ibrahim Labis	1 (0.2)	

**Table 4 children-09-01883-t004:** Participants’ results to different questionnaire.

Scale	Level (Threshold)	Number (*n* = 500)	Percentage of Frequency (%)	Mean (SD)
Internet Addiction Test	Normal Mild Moderate Severe	173 179 138 10	34.6 35.8 27.6 2.0	39.57 (18.928)
Depression Anxiety Stress Scale—21				
Depression	Normal Mild Moderate Severe Extremely Severe	176 48 114 90 72	35.2 9.6 22.8 18.0 14.4	14.87 (10.778)
Anxiety	Normal Mild Moderate Severe Extremely Severe	110 24 115 69 182	22.0 4.8 23.0 13.8 36.4	16.15 (10.704)
Stress	Normal Mild Moderate Severe Extremely Severe	243 77 83 69 28	48.6 15.4 16.6 13.8 5.6	15.79 (10.187)
TIPI				
Openness to Experience Conscientiousness Extraversion Agreeableness Emotional Stability				4.62 (1.196) 4.34 (1.173) 3.78 (1.218) 4.73 (1.066) 4.28 (1.257)

**Table 5 children-09-01883-t005:** Results of indirect effect.

Relationship	β	*t*	*p*	VIF	Supported
Internet Addiction → Conscientiousness → Emotional Distress	0.012	1.133	0.257	1.29	No
Internet Addiction → Openness to Experience → Emotional Distress	0.001	0.005	0.996	1.16	No
Internet Addiction → Agreeableness → Emotional Distress	0.002	0.297	0.766	1.21	No
Internet Addiction → Emotional Stability → Emotional Distress	0.087	4.416	0.00	1.16	Yes
Internet Addiction → Extroversion → Emotional Distress	0.005	0.509	0.611	1.07	No

## Data Availability

Data are available from the corresponding author with the permission of Universiti Malaysia Sabah.

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
