# Peer review of "Relationship between Internet Addiction, Personality Factors, and Emotional Distress among Adolescents in Malaysia"

_children, 2022, doi:10.3390/children9121883_

Round 1
Reviewer 1 Report
Chou Fu et al aimed to determine the prevalence of Internet addiction and emotional distress (depression, stress, and anxiety) among adolescents in Malaysia.
It is an interesting article but there are some shortcompings that need attention. I would like to make some suggestions and contributions to the present manuscript. The authors need to check that their manuscript complies with STROBE guidelines for observational research: https://www.equator-network.org/reporting-guidelines/strobe/ A statement needs to be added to the manuscript confirming the same.
Introduction
The purpose of the article is clearly presented. Overall this is well-written and puts the study in context with reference to up-to-date and relevant literature, however, strengthening the rationale is likely to improve the manuscript.
Although the authors have included relevant references, it is still too brief an introduction. It is recommended that the authors include more information and data on the prevalence of Internet addiction, as well as data on the prevalence of mental disorders (depression, anxiety and stress) during the COVID-19 pandemic context.
Methods
The investigation has been scientifically and methodically conducted. Authors used good language and statistical methods
The authors should provide information on the study population of the seven schools selected for the study. For example: School A (N=XX students), etc.
The authors should develop a participant selection flow chart. This will allow you to know the total number of students excluded from the study.
The authors should specify more information about the sampling and sampling techniques used for this research.
The authors should provide more information on the inclusion criteria for your study.
The authors should provide more detail on the procedures used for their research.
Results
The statistical analysis was highly rigid. The statistical analysis and its interpretation are appropriate
The authors should be uniform when presenting the rounding of decimals.
I suggest adding an image showing the distribution of questionnaire responses for the variables of interest (Internet addiction and mental health).
Discussion
The discussion is very limited and the literature used has been documented during pre-pandemic period, studies with similar/contrary findings conducted during pandemic by COVID-19 should be discussed with studies with similar/contrary findings conducted during pandemic by COVID-19. Findings on 1) internet addiction, 2) depression, anxiety and stress and 3) mediating role of personality factors should be discussed in a separate paragraph.
Authors should substantially improve the discussion. In addition, in the limitations described, it is not enough to mention each one of them, but they should mention what was the potential solution to deal with these limitations and explain why their presence does not invalidate their study results.
Conclusions
Conclusions are logically valid and justified by evidence about the main theme proposed.
With these changes, readers will be able to fully appreciate the potential clinical significance of the findings and future research directions. I hope that these proposed modifications will serve to improve the manuscript.
Author Response
Dear Reviewer 1, we are grateful for your consideration of this manuscript, and we also very much appreciate your suggestions, which have been very helpful in improving the manuscript. All the comments we received on this manuscript have been taken into account in improving the quality.

Reviewer 2 Report
This is an interesting study examining the relationship between internet addiction, personality factors, and emotional distress among adolescents in Malaysia. I have several major concerns regarding the manuscript:
1. First, I am concerned with the use of personality traits as mediator of the relationship between internet addiction and emotional distress. There is no strong theoretical reason to expect personality to be malleable and explain mechanism underlying internet addiction and emotional distress. As a trait variable, it is not logical to expect big five personality traits as a mediator. Due to the cross-sectional design of the study, the mediation analyses are misleading. The significant mediating effect of emotional stability can be simply a risk factor rather than effect of internet addiction. I would strongly suggest the author to remove personality traits as their mediators. If the authors insist to include them, personality traits should be predictors or moderators rather than mediators.
MacKinnon, D. P., Fairchild, A. J., & Fritz, M. S. (2007). Mediation analysis. Annual Review of Psychology, 58, 593.
2. The sample size and participants' characteristics should be moved to participant section in the Method.
3. There is a need for the authors to supplement more information about their sample. For example, it will be important for the authors to provide more information on how the sample was recruited. Inclusion and exclusion criteria should be elaborated. The authors should also comment on whether the sample is representative
4. More information is necessary regarding the validity of the internet addiction test used in the current study.
5. Internal consistency of the scales used in the study seems to be low for most. This should be acknowledged in the limitation section.
6. It will be important for the authors to mention how missing data was treated in the analysis. If listwise deletion was used, it will be good to provide a brief justification. Relevant paper:
Newman, D. A. (2014). Missing data: Five practical guidelines. Organizational Research Methods, 17(4), 372-411.
7. Another important point to discuss in the Discussion section is the possibility of reverse causation which is a common limitation in the literature. While the authors suggested that emotional distress were likely due to internet addiction, it is equally plausible that internet addiction is the antecedent of emotional distress. This problem related to reverse causation should be discussed in details to provide a more balanced interpretation of the result. This is a critical point to discuss. Please see the following relevant paper: Quek, F. Y., Tng, G. Y., & Yong, J. C. (2021). Does social media use increase depressive symptoms? A reverse causation perspective. Frontiers in Psychiatry, 12, 335.
8. The model fit of the structural equation model should be reported and assessed. West, S. G., Taylor, A. B., & Wu, W. (2012). Model fit and model selection in structural equation modeling. Handbook of structural equation modeling, 1, 209-231.
Author Response
Dear Reviewer 2, we are grateful for your consideration of this manuscript, and we also very much appreciate your suggestions, which have been very helpful in improving the manuscript. All the comments we received on this manuscript have been taken into account in improving the quality.

Round 2
Reviewer 1 Report
Dear Editor
The authors have done a good job in addressing my comments. I have no other input.
Reviewer 2 Report
The authors have addressed all my comments well. I appreciate all their efforts. Well done.